# Physiological Performance of Poplar and Willow Clones Growing on Metal-Contaminated Landfills

**DOI:** 10.3390/plants14111705

**Published:** 2025-06-03

**Authors:** Lazar Kesić, Branislav Kovačević, Marina Milović, Dragica Stanković, Marko Ilić, Leopold Poljaković-Pajnik, Saša Pekeč, Saša Orlović

**Affiliations:** 1Institute of Lowland Forestry and Environment, University of Novi Sad, Antona Cehova 13d, 21102 Novi Sad, Serbia; branek@uns.ac.rs (B.K.); katanicm@uns.ac.rs (M.M.); marko.ilic@uns.ac.rs (M.I.); leopoldpp@uns.ac.rs (L.P.-P.); pekecs@uns.ac.rs (S.P.); sasao@uns.ac.rs (S.O.); 2Institute for Multidisciplinary Research, University of Belgrade, Kneza Višeslava 1, 11030 Belgrade, Serbia; dstankovic@imsi.bg.ac.rs

**Keywords:** landfill, leaf gas exchange, willows, poplars, biomass

## Abstract

This study evaluated the physiological responses and biomass production of selected poplar and willow clones cultivated in form of phytoremediation buffer plantations on landfills in Vinča (near Belgrade) and Novi Sad, Serbia. Key parameters assessed included net photosynthesis (A), transpiration (E), stomatal conductance (g_s_), and water use efficiency (WUE). Results indicated a significant Clone × Site interaction for net photosynthesis, suggesting environmental-specific clone responses. Transpiration and stomatal conductance exhibited site-stable expression between sites, implying conservative traits or similar hydrological conditions during measurements. Particularly, total site values for physiological parameters were higher at the Novi Sad site, likely due to continuous access of plants to groundwater. The weak correlation between WUE and biomass production suggests that favorable water conditions at both sites diminished the importance of water use efficiency for biomass accumulation. Poplar clone S1-8 exhibited the highest biomass production and leaf-level gas exchange traits (A, E, g_s_, WUE), reflecting a fast-growth strategy through increased gas exchange. This clone’s consistent productivity across sites classifies it as a generalist, while willow clone 378 and poplar clone 135/81, with significantly higher biomasses at the Novi Sad site than at the Vinča site, can be considered as specialists. Use of both generalist and specialist clones in multiclonal plantations may enhance phytoremediation and biomass production stability across variable sites. These findings underscore the importance of selecting appropriate clones for phytoremediation on landfills and on contaminated lands in general.

## 1. Introduction

The rapid expansion of the human population, coupled with intensive industrial and economic development over the past decades, has led to significant environmental degradation, contamination, and climate change [1]. Landfills are particularly concerning, as they release leachate loaded with pollutants such as toxic organic compounds and heavy metals, which can severely impact nearby water basins [2]. Landfills present unique challenges for vegetation due to factors such as soil contamination, limited nutrient availability, and altered hydrological conditions [3,4]. Despite these challenges, certain tree species, notably poplars (*Populus* spp.) and willows (*Salix* spp.), have demonstrated remarkable adaptability, making them prime candidates for phytoremediation projects on such sites. Many contaminants pose substantial threats to both ecosystem integrity and human health, particularly due to the high concentrations of heavy metals such as lead (Pb), cadmium (Cd), and mercury (Hg), which can accumulate in soil and water, disrupt biological processes, and enter the food chain, leading to long-term ecological degradation and serious health risks including neurotoxicity, organ damage, and developmental disorders [5]. However, physiological performance under such conditions is a key determinant of their success, influencing growth dynamics, biomass production, and overall ecosystem benefits [6].

Photosynthesis is the fundamental physiological process driving carbon assimilation and growth in trees. Its efficiency is influenced by environmental factors such as soil composition, water availability, and atmospheric conditions [7]. In landfill environments, where soil fertility is low and contaminants may be present, photosynthetic capacity can be affected, leading to variations in carbon fixation and biomass accumulation [6]. Additionally, transpiration (E) and stomatal conductance (g_s_) regulate water loss and CO₂ exchange, impacting the overall physiological status of plant [8]. Stomatal conductance plays a crucial role in determining how efficiently plants utilize available water, especially in landfill ecosystems where water retention properties may be poor [9]. The ability of different poplar and willow clones to optimize their water use efficiency (WUE) is of particular importance, as it affects their survival and productivity in such stress conditions [10].

One of the key indicators of the success of afforestation on landfills is dry shoot biomass production. Biomass accumulation is not only an indicator of tree productivity but also of the species’ ability to capture and store carbon [11]. Given the increasing global emphasis on mitigating climate change, assessing the total accumulated CO₂ equivalent provides insights into the potential of these species to serve as carbon sinks in degraded environments [12,13]. By selecting clones that demonstrate superior physiological performance, characterized by high net photosynthesis, optimal transpiration rates, and efficient stomatal regulation, landfill afforestation projects can be optimized for maximum environmental benefits [14].

We hypothesize that clone-specific physiological adaptation to landfill conditions will differentially influence biomass production depending on water access and soil properties. The main objectives of the study were: (1) to compare physiological and growth performance, based on gas exchange parameters and biomass accumulation between willow and poplar clones, and (2) to investigate the interaction between physiological and growth responses of examine clones and the specific environmental conditions of landfill sites.

## 2. Results

### 2.1. Heavy Metal Content in Plant Material

Duncan’s multiple range test indicated statistically significant differences between the landfill sites in terms of barium, cadmium, copper, and zinc concentrations in the aboveground biomass of the plants studied, with consistently higher accumulation observed at the Novi Sad site (Table 1). When comparing the total uptake among clones, a significant difference was found in barium accumulation between clones 378 and S1-8, while clone PE19/66 exhibited a markedly higher accumulation of iron compared to the others. Clonal differentiation was particularly distinct with respect to cadmium and zinc uptake. Notably, white willow clone 380 demonstrated the highest accumulation of cadmium, with white willow clones, in general, showing superior uptake capacity for this metal. This pattern was even more pronounced for zinc, where white willow clones clearly exhibited dominant accumulation potential (Table 1).

### 2.2. Results of Gas Exchange Parameters

When analyzing the mean values of the examined physiological parameters across clones and locations, it is evident that significant differences exist among the clones, with higher values for all physiological traits recorded at the Novi Sad landfill site. In our study, the highest net photosynthesis rate (A) and water use efficiency (WUE) were observed in clones 378 and I-214. Clone Pannonia exhibited the lowest transpiration rate (E), whereas clones S1-8 and 107/65/9 had the highest stomatal conductance (g*_s_*) (Table 2). However, despite its high biomass production, clone S1-8 showed a lower net photosynthesis rate compared to the other clones. Notably, at the “Vinča” landfill, clone S1-8 demonstrated the highest biomass production and carbon accumulation, along with high net photosynthesis and water use efficiency, but also exhibited high transpiration and stomatal conductance. However, at the “Novi Sad” landfill, this clone also led in biomass production and carbon accumulation, displaying high stomatal conductance and transpiration rates, but achieved a relatively moderate net photosynthesis rate and water use efficiency comparing to other clones (Table 2).

Overall, the clones responded positively to the environmental conditions at the “Novi Sad” landfill, but there was considerable variability in dry biomass per unit area (m*_P_*). However, only the *S. alba* clone 378 and the *P. deltoides* clone 135/81 exhibited statistically significant differences in this parameter between the two locations. Specifically, these two clones achieved significantly higher dry biomass values at the Novi Sad site compared to the Belgrade site (Table 1). In total, the *P. deltoides* clone S1-8 dominated also in terms of biomass production, showing no significant difference in biomass yield between two sites. At the Novi Sad site, clone S1-8 recorded the highest dry biomass per unit area (16,634.33 ± 2406.81 kg ha⁻^1^), significantly outperforming the *P. deltoides* clone PE19/66 and the *P. x euramericana* clones I-214 and Pannonia, as well as the *S. alba* clone 107/65-9. Additionally, at the same site, the *S. alba* clone 378 exhibited high aboveground biomass production (15,015.64 ± 2406.81 kg ha⁻^1^), which was significantly greater than that of PE19/66 and Pannonia (Table 2).

The first two principal components accounted for 93.60% of the total observed variation, indicating a strong ability of these two principal components to summarize and present the dataset’s overall variability. They were selected according to the Keiser’s role, as their eigenvalues were higher than 1.0. The total CO_2_ equivalent sequestration in woody biomass was not included in this analysis because it is calculated from m*_P_* and their correlation is 1.0, meaning that it does not contribute any additional data to the total variation. Based on the loadings of the first two principal components with the examined parameters, the majority of the measured parameters showed their highest correlations with the first principal component (PC1), including stomatal conductance (g*_s_*), transpiration rate (E), net photosynthesis and dry shoot biomass (m*_P_*). In contrast, parameter water use efficiency (WUE) exhibited its highest loading with the second principal component (PC2). The fact that g*_s_*, E and A are in the same group with m*_P_* suggest high significance of these parameters for biomass production in this study, which is not the case with the water use efficiency (Figure 1).

The first axis (PC1) shows a clear differentiation of clones between the two landfills, where most of the interaction treatments at the Vinča site are on the positive side, and most of the interaction treatments at the Novi Sad site are on the negative side of the axis. All traits included in this analysis are in negative correlation with the first principal component (Figure 2); therefore, high negative PCA1 values of some treatment suggest high values of its leaf gas exchange and biomass yield characters. Thus, negative PCA1 values for NS treatments suggests superiority of the performance of examined clones on NS site comparing to the Vinča site, for which PCA1 values of examined clones were mostly positive. In this context, it could be said that examined clones achieved higher transpiration rate, photosynthetic rate and stomatal conductance, as well as biomass yield on NS site than on BG site, which is in concordance with the results of Duncan test. In contrast, the second axis (PC2) distinctly separates the treatments of clones on the Novi Sad landfill, namely clones 380, 135/81 and S1-8 from I-214, 107/65/9, and 378. The distinction between two groups of clones on BG site was not so clear. This differentiation indicates that specific environmental conditions or site-specific factors are driving the variation captured by PC2, highlighting stronger influence of location-dependent variables on the studied clones on NS site (Figure 2). Considering the fact that only WUE achieved its highest loading with the second principal component, the differentiation is closely related to the differentiation between clones by WUE. Considering this fact, clones I-214, 107/65-9 and 378 used water more efficiently, than 380, 135/81 and S1-8 at the “Novi Sad” landfill, while this distinction was not clear at the “Vinča” site.

## 3. Discussion

Physiological parameters of leaf gas exchange, such as net photosynthesis, transpiration, stomatal conductance, and water use efficiency, represent reliable indicators of plant vitality and productivity, particularly under stress conditions that prevail at contaminated sites like landfills [15]. This study focused on the physiological responses of various poplar and willow clones exposed to contaminated substrates from landfills in Belgrade (Vinča) and Novi Sad, with the aim of evaluating their adaptability, and potential for phytoremediation in the conditions of examined landfills.

The concentration of heavy metals in plant biomass is a critical criterion in the selection of willow and poplar clones for phytoremediation purposes, particularly in the context of contaminated land such as municipal landfills, mining sites, or industrial zones. Clones that accumulate higher levels of cadmium, zinc, or copper in aboveground biomass are especially valuable for phytoextraction in contaminated environments such as landfills or industrial sites [16,17]. For instance, a recent field study demonstrated that the willow clone *Salix smithiana* (S2) achieved the highest biomass yield and effectively removed cadmium and zinc from contaminated soils. This clone removed up to 9.07% of Cd and 3.43% of Zn from the topsoil layer (0–20 cm), indicating its strong phytoextraction potential. In contrast, the poplar clone *Populus Max-4* (P1) showed higher lead (Pb) accumulation over an 8-year rotation, though overall Pb removal was limited due to its low mobility in soil [18]. Another study from 2025 assessed five poplar and two white willow clones in pot trials using soils from landfills near Belgrade and Novi Sad. The white willow clone 107/65-9 exhibited the highest accumulation of multiple heavy metals, including cadmium, chromium (Cr), iron (Fe), nickel (Ni), and lead, acting as a generalist. In contrast, poplar clones displayed specialization, with certain clones accumulating specific metals more effectively [19]. Clones with efficient uptake and tolerance mechanisms enhance the sustainability of remediation efforts.

The results indicated that the Clone × Site interaction was statistically significant for net photosynthesis, suggesting that response to environmental conditions may be clone specific, and that not all physiological response parameters necessarily change proportionally across locations. The clone means for transpiration and stomatal conductance did not show significant differences between sites for any examined clone, which may indicate the conservative nature of these traits or similar hydrological conditions during the measurement period. However, when observing the total landfill mean values of all physiological parameters, it was found that the values were significantly higher at the Novi Sad landfill site.

This may be attributed to more favorable habitat conditions that occur in Novi Sad site, presumably continuous access to groundwater, which enhances soil moisture availability and promotes stability in the plants’ water regime. A stable and readily accessible water supply plays a crucial role in maintaining cell turgor, enabling efficient stomatal regulation, and sustaining photosynthetic machinery under potential stress conditions [20,21,22]. In environments where water availability is not a limiting factor, plants are less likely to experience hydraulic stress, allowing them to maintain open stomata for longer periods [23]. This facilitates greater carbon dioxide (CO₂) uptake, leading to increased rates of photosynthesis [24,25,26].

Furthermore, continuous soil moisture enhances nutrient solubility and uptake, indirectly supporting metabolic processes such as chlorophyll biosynthesis and Rubisco activity, which are fundamental to maintaining high photosynthetic efficiency [27,28]. The synergistic effect of stable hydration and nutrient availability can improve overall plant physiological status, resulting in higher productivity, especially in species with high biomass potential like poplars and willows [29]. Thus, in our study that examined growth and physiology of poplar and willow clones cultivated in landfill soils, we found that consistent moisture and nutrient availability were crucial for optimal biomass production. The research indicated that examined species could effectively establish and thrive in landfill conditions, if provided with sufficient water and nutrients. This underscores the importance of managing soil moisture and fertility to support the metabolic processes essential for growth and photosynthesis in trees [30]. Furthermore, research conducted by Rogers et al. [31] on willow growth in response to nutrients and moisture on a clay landfill demonstrated that water stress reduced stem biomass production by 26–37% and resulted in higher root-to-stem ratios. This highlights the significant impact of water availability on biomass yield and the importance of maintaining adequate hydration for optimal growth. Our results are in concordance with these findings, which demonstrated higher biomass production at “Novi Sad” landfill, where steady availability of groundwater was provided, than at “Vinča” landfill, where the soil nutrient content was higher, but plants had no contact with groundwater leading to poor water regime of plants.

Water-abundant conditions are particularly beneficial in contaminated substrates, like those found at landfill sites, where water stress can exacerbate the toxic effects of heavy metals or other pollutants on photosynthetic performance and gas exchange [32]. Additionally, Nikolić et al. [33] identified a significant positive correlation between photosynthesis and biomass under heavy metal-contaminated soil culture, supporting the hypothesis that photosynthetic activity is a reliable indicator of productivity under stress. Therefore, the increased photosynthetic activity observed at the Novi Sad landfill site may be a reflection of both improved water status and reduced physiological stress due to more stable environmental conditions. Water use efficiency is considered to be a key physiological indicator for evaluating plant productivity, particularly under conditions of water limitation [14]. It represents the ratio between carbon assimilation through photosynthesis and water loss via transpiration. In the context of landfill sites afforestation, understanding WUE is crucial for the selection and management of tree species capable of adapting to the often harsh and variable environmental conditions [34]. However, findings from our study indicate that no significant correlation was observed between WUE and biomass production, suggesting that other factors instead of the efficiency of water use may play a more important role in determining growth performance under harsh conditions [5]. This may be attributed to the fact that water conditions were favorable at both sites—due to groundwater contact in Novi Sad and regular irrigation in Belgrade—making differences in WUE among clones less pronounced compared to other parameters such as g**_s_** and E [6]. In that sense, the poor relationship of WUE with m**_P_** that was found in our study indicates that the abundance of water diminished the importance of WUE for the accumulation of biomass, suggesting that increased transpiration efficiency under non-limiting conditions leads to high biomass production.

Indeed, the highest values of net photosynthesis and water use efficiency were recorded in clones 378 and I-214, indicating their good adaptive potential in relation to water availability and the presence of contaminants. Clone S1-8 exhibited high stomatal conductance and transpiration rate which typically facilitate greater CO₂ uptake and support high photosynthetic activity [35]. However, despite this, its water use efficiency remained moderate, suggesting a potential trade-off between carbon gain and water conservation. This may indicate that S1-8 prioritizes carbon assimilation under the given conditions, possibly as an adaptive strategy in environments where water is not a limiting factor. The high g**_s_** and E could lead to increased water loss through transpiration, thereby reducing intrinsic WUE (the ratio of photosynthesis to stomatal conductance) or instantaneous WUE (the ratio of photosynthesis to transpiration). Such a physiological profile is often observed in fast-growing genotypes that maximize growth under favorable conditions, even at the expense of lower water-use efficiency. This trade-off highlights the importance of balancing productivity and resource-use strategies in clone selection for phytoremediation or biomass production, especially under variable or stress-prone environments [36]. Increased g**_s_** enables longer stomatal opening duration, and thus prolonged CO₂ assimilation, which may explain the good productivity of this clone even under less favorable conditions that occurred at the “Vinča” site. Clone S1-8 stood out as the most productive clone at both landfills, classifying it as a generalist—capable of maintaining high productivity across different environments. Conversely, clones 378 and 135/81 achieved significantly higher values of dry biomass per unit area at the Novi Sad site compared to Vinča site, which positions them as ecological specialists for more favorable conditions, such as those with a stable water regime. A comparison with the available literature confirms that these findings are in line with the study by Kesić et al. [14], where clones S1-8 and PE4/68 exhibited the highest physiological parameter values in pot culture.

It is important to note that biomass production varied among clones, and that clone S1-8 achieved the highest aboveground biomass at both sites—16,634 kg ha^−1^ in Novi Sad and slightly less in Belgrade. Similar values were recorded for clone 378, identifying it as a promising candidate for phytoremediation programs. According to Zalesny et al. [2], aboveground biomass values of poplars treated with landfill leachate in the U.S. ranged from 510 to 2500 kg ha^−1^, which are significantly lower compared to our findings. Furthermore, Zalesny et al. [37] reported that the global average annual biomass increment for poplars is 11.2 Mg ha^−1^ yr^−1^, and 12.3 Mg ha^−1^ yr^−1^ in North America. Our results, with an average value of 13,210 kg ha^−1^ for clone S1-8, indicate a high biomass production potential of this clone even under landfill substrate conditions.

This classification of clones based on their performance across different environments suggests a distinction in ecological specialization: generalist clones (such as S1-8) demonstrate suitability for a wide range of site conditions, while specialist clones (e.g., 378 and 135/81) perform better under more favorable and stable environmental conditions. Utilizing a combination of both types of genotypes may enhance the overall stability of phytoremediation functions and biomass production across variable sites, as recommended by Zalesny et al. [38].

## 4. Materials and Methods

### 4.1. Experimental Design and Site Properties

In our study, one-year-old rootless whips were planted at a depth of approximately 80 cm, with a spacing of 1.5 × 1.5 m, resulting in a planting density of 4444 plants per hectare. The experiments were conducted at two locations in the field trials established in the winter 2023/2024: the “Novi Sad” landfill (N 45°18′, E 19°50′, altitude 75 m), and the “Vinča” landfill near Belgrade (N 44°47′, E 20°36′, altitude 99 m). Plant survival was highly successful at both sites, with a survival rate of examined clones consistently exceeding 95%.

The soil at both sites is classified as anthropogenic. At the “Vinča” landfill, the soil had a heavier texture (sandy clay loam) compared to the “Novi Sad” landfill (loamy sand) [39]. It also contained a slightly higher humus content in the deeper soil layers (below 40 cm) and significantly higher levels of nitrogen, phosphorus, and potassium. Soil samples for analysis were collected from the central part of the plantation. At each site, one composite sample was taken at three different depths, providing a representative profile of the soil layers. This sampling approach was chosen to accurately reflect the vertical distribution of soil properties relevant to plant growth and to capture potential variations in nutrient content and contamination levels across different soil horizons. All analyses were conducted at the Laboratory of Soil Science, Institute of Lowland Forestry and Environment in Novi Sad, following standardized methodologies in accordance with quality assurance and control procedures outlined by [40], as well as the guidelines provided in the ICP Forests Manual for soil sampling and analysis [41] (Table 3 and Table 4).

At the “Novi Sad” site, the groundwater depth fluctuated between 65 and 113 cm during the growing season, meaning that the plants had direct access to groundwater or were within the capillary rise zone, which ensured a stable water supply for the plant. In contrast, at the “Vinča” landfill, groundwater was neither directly available nor accessible through capillary action, making precipitation the only natural water source. As a result, the plantation has been regularly irrigated once every two weeks from May until August.

### 4.2. Plants Material

The selection of clones was carried out based on the results of phyto-recurrent selection in pot trials, as described by Kesić et al. [14] and Kovačević et al. [19]. The testing process was conducted in three successive cycles, focusing on a comprehensive evaluation of morphometric traits, biomass properties, and physiological leaf parameters. These assessments were performed on substrates specifically formed from the soil collected at the “Novi Sad” and “Vinča” landfills to simulate actual growing conditions. The experiment was set up in three replications. Each replication included the same set of clones arranged in blocks. Within each block, there were nine individual plants (arranged in a 3 × 3 pattern). This experimental design ensured uniform distribution and replication of genetic material, allowing for statistically robust comparisons of physiological performance across clones and locations.

A total of 34 clones were analyzed, encompassing three key species: eastern cottonwood (*Populus deltoides* Bartr.), Euroamerican poplar (*Populus × euramericana* Dode (Guinier)), and white willow (*Salix alba* L.). Based on their adaptability and growth performance on these landfill-derived substrates, six clones were recommended for plantation establishment. The selection criteria included not only their ability to thrive in the given soil conditions but also their potential for biomass production and ecological restoration.

In addition to the newly selected clones, several widely used and well-adapted reference clones were included as controls to provide a performance benchmark. These control clones included the Euroamerican poplar varieties I-214 and Pannonia, both of which are extensively cultivated due to their proven adaptability and productivity. Furthermore, the white willow clone 378, officially registered as 5/3-378, was selected for the trial at “Novi Sad” but it was also incorporated into the trial at “Vinča” landfill, as a standard reference. The inclusion of these control clones allowed for a more precise comparison of growth characteristics and resilience under the tested landfill conditions, ensuring that the recommended clones demonstrate significant potential for successful establishment and sustainable biomass production (Table 5).

### 4.3. Assessment of Carbon and Heavy Metal Content in Aboveground Woody Biomass

At the end of the first growing season, in January and February of 2025, the height and diameter at 10 cm above the soil surface were measured for each plant in the experiment. Additionally, plant survival was assessed through the end of the growing season. For each clone plot within the block, the total biomass of the above-ground part of the plants was measured. A sample was taken from the middle section of the stem (cutting) to determine the moisture content of the above-ground biomass. Based on these moisture content measurements, the dry biomass of the above-ground part was calculated for each clone plot within the block. This process aimed to provide a comprehensive evaluation of the growth and survival rates of the clones, alongside an accurate estimate of their above-ground biomass production.

The carbon accumulated in woody biomass of the clones as well as equivalent CO_2_ were calculated based on following formulae:(i)the amount of accumulated carbon in the above-ground parts of the plant per area unit (C_tot_ [kg ha^−1^]):C_tot_ = C × m_P_
where the coefficient C represents the total carbon partition of the dry woody biomass (C = 0.4925 in the dry biomass of poplar, and C = 0.4905 in the dry biomass of willow) [42], and m**_P_** denotes the mass of dry woody biomass per area unit [kg ha^−1^];
(ii)and the amount of carbon dioxide equivalent to the amount of accumulated carbon in the above-ground parts of the plant per unit area (CO_2_ eq [kg ha^−1^]):
CO_2_ eq = 3.67 × C_tot_
where the coefficient 3.67 represents the ratio of the molecular weight of CO_2_ to the atomic weight of C, and C_tot_ denotes the carbon accumulated in the above-ground woody biomass per area unit [kg ha^−1^].

For the analysis of heavy metal content, aboveground wood biomass was used. The samples were oven-dried, homogenized, and ground, then digested in aqua regia (a mixture of concentrated hydrochloric and nitric acid in a 3:1 ratio). The resulting extract was filtered and made up to a final volume of 50 mL. The quantification of heavy metals was performed using an ICP-OES spectrometer VISTA-PRO (Varian Australia Pty. Ltd., Melbourne, VIC, Australia). The radiofrequency power was set to 1100 W to ensure efficient plasma generation, with a plasma gas flow rate of 15.0 L min^−1^ and an auxiliary gas flow rate of 1.50 L min^−1^ to support plasma stability. Measurements were carried out in three replicates, following the methodology described by Kebert et al. [43].

### 4.4. Assessment of Leaf Gas Exchange

To assess the physiological performance of different clones, key leaf gas exchange parameters were measured, including net photosynthesis (A, [μmol m^−2^ s^−1^]), transpiration rate (E, [mmol m^−2^ s^−1^]), stomatal conductance (gₛ, [mmol m⁻^2^ s^−1^]), and water use efficiency (WUE, [mmol mol^−1^]). These parameters provide valuable insights into the photosynthetic activity and water regulation efficiency of the plants studied.

The measurements were conducted using a CIRAS-3 portable photosynthesis system (Amesbury, MA, USA) during the morning hours, between 9:00 a.m. and 11:00 a.m., to capture optimal physiological activity. Data collection was carried out over two consecutive days, with measurements taken at the “Vinča” landfill on 18 July 2024, and at the “Novi Sad” landfill on 19 July 2024. To ensure consistency in environmental conditions, the measurements were performed under a standardized photosynthetically active radiation (PAR) level of 1000 μmol m^−2^ s^−1^ and an atmospheric CO₂ concentration of 390 μmol mol^−1^.

For each experimental treatment, three random plants from each clone were measured within each block. From each plant, gas exchange parameters were recorded from a single, fully expanded leaf, with three to five individual readings taken per leaf. This approach ensured a representative dataset for evaluating the physiological responses of different clones under the given experimental conditions.

### 4.5. Statistical Analysis

The data analysis was conducted using a two-way factorial analysis of variance (ANOVA), followed by Duncan’s post hoc test, applied for multiple comparisons of treatments. The test aimed to identify significant differences (1) between treatments, (2) between clones, and (3) in the interaction between clones and treatments for *p* < 0.05.

Additionally, principal component analysis (PCA) was used to explore the relationships between treatments at the interaction level, based on the first two principal components. PCA was also employed to examine the relationships between the measured parameters. These parameters were grouped according to their highest factor loadings with the first two principal components. All statistical analyses were performed using the STATISTICA 14.0.0 software package [44].

## 5. Conclusions

In conclusion, the findings of this study highlight the significant variability in physiological responses and biomass production among different poplar and willow clones grown on contaminated landfill sites, emphasizing their potential for use in phytoremediation and biomass production. Under an abundant supply of water, gas exchange traits (g**_s_**, E, A) were stronger predictors of biomass yield than intrinsic water use efficiency (WUE). Clone S1-8 emerged as a highly productive generalist, capable of maintaining superior physiological performance and biomass yield across diverse environmental conditions, while clones such as 378 and 135/81 showed a higher degree of specialization, performing best under more stable and water-rich conditions. These insights underscore the importance of selecting appropriate genotypes based on site-specific characteristics to optimize both phytoremediation and biomass output. By strategically combining generalist and specialist clones, it is possible to enhance the resilience and efficiency of phytoremediation systems in degraded and heterogeneous landscapes, thereby contributing to sustainable land management and climate mitigation efforts.

## Figures and Tables

**Figure 1 plants-14-01705-f001:**
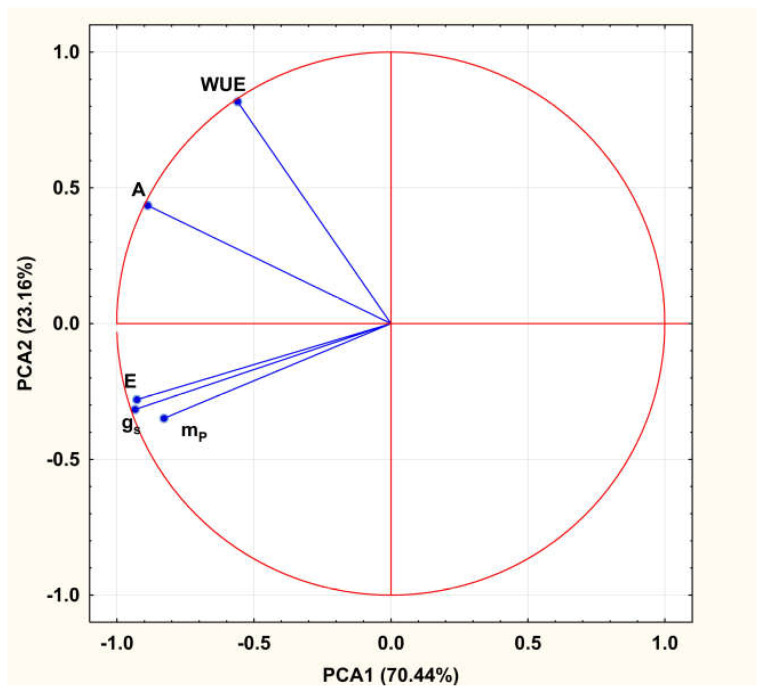
Factor loadings of original parameters with the first two principal components.

**Figure 2 plants-14-01705-f002:**
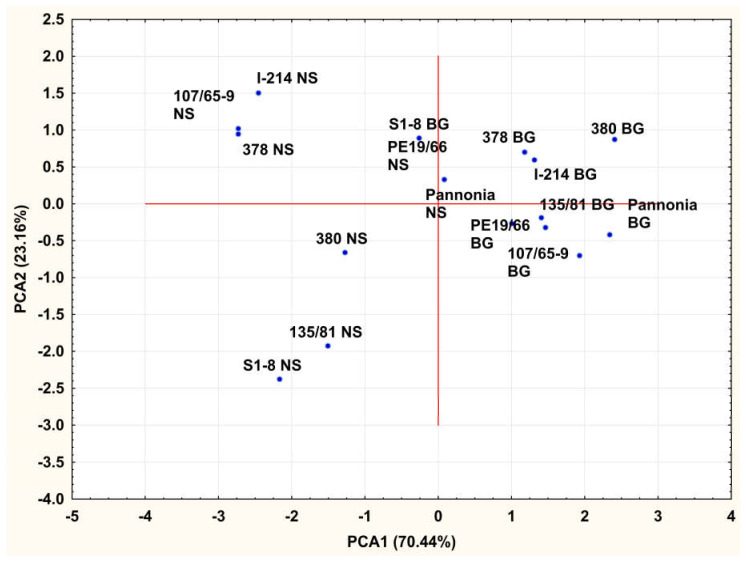
Relationship between Clone × Landfill interaction treatments according to factor scores of the first two principal components (BG stands for Belgrade landfill substrate and NS for Novi Sad landfill substrate) based on leaf gas exchange traits and biomass yield. Clones 107/65-9, 378 and 380 are willow, and the rest are poplar clones.

**Table 1 plants-14-01705-t001:** Results of Duncan’s test for the content of examined heavy metals in the aboveground biomass [mg kg^−1^ dry weight] of the studied poplar and willow clones at landfill sites in Novi Sad (NS) and Vinča (BG).

P/S ^(1)^	Clone	Landfill ^(2)^	As ^(3)^	Ba	Cd	Cr	Cu	Fe	Ni	Zn
	Means at the level of interaction clone x landfill									
S	378	BG	1.392 ^a (4)^	4.811 ^bc^	0.546 ^bc^	0.435 ^b^	8.672 ^b^	34.927 ^b^	1.205 ^a^	39.235 ^cde^
		NS	0.646 ^a^	8.780 ^a^	0.544 ^bc^	0.707 ^ab^	17.813 ^a^	37.052 ^b^	2.507 ^a^	56.265 ^a^
S	380	BG	1.433 ^a^	3.158 ^c^	0.631 ^ab^	0.830 ^ab^	10.089 ^b^	48.507 ^b^	1.211 ^a^	53.466 ^ab^
		NS	0.143 ^a^	5.652 ^abc^	0.805 ^a^	0.746 ^ab^	17.104 ^a^	41.056 ^b^	0.980 ^a^	63.796 ^a^
S	107/65/9	BG	1.023 ^a^	3.799 ^bc^	0.462 ^bcd^	0.533 ^ab^	9.423 ^b^	41.785 ^b^	1.366 ^a^	32.676 ^def^
		NS	0.814 ^a^	7.029 ^ab^	0.548 ^bc^	0.829 ^ab^	17.765 ^a^	36.746 ^b^	1.563 ^a^	55.962 ^a^
P	135/81	BG	1.525 ^a^	4.604 ^bc^	0.429 ^cd^	0.491 ^ab^	9.556 ^b^	38.732 ^b^	1.464 ^a^	25.774 ^fg^
		NS	0.564 ^a^	6.236 ^abc^	0.519 ^bcd^	0.549 ^ab^	15.162 ^a^	35.687 ^b^	1.319 ^a^	38.716 ^cde^
P	I-214	BG	0.000 ^a^	4.030 ^bc^	0.397 ^cd^	0.498 ^ab^	8.856 ^b^	35.050 ^b^	2.446 ^a^	17.354 ^g^
		NS	1.891 ^a^	4.522 ^bc^	0.386 ^cd^	0.534 ^ab^	16.848 ^a^	28.077 ^b^	1.387 ^a^	30.968 ^ef^
P	Pannonia	BG	0.000 ^a^	4.597 ^bc^	0.348 ^cd^	0.491 ^ab^	9.267 ^b^	36.030 ^b^	1.140 ^a^	22.623 ^fg^
		NS	1.234 ^a^	4.713 ^bc^	0.534 ^bcd^	0.459 ^b^	18.273 ^a^	41.055 ^b^	0.922 ^a^	42.442 ^cd^
P	PE19/66	BG	0.746 ^a^	3.580 ^bc^	0.388 ^cd^	1.241 ^a^	10.367 ^b^	82.723 ^a^	1.737 ^a^	27.125 ^fg^
		NS	1.731 ^a^	6.239 ^abc^	0.644 ^ab^	0.882 ^ab^	18.010 ^a^	44.100 ^b^	1.445 ^a^	41.155 ^cde^
P	S1-8	BG	0.893 ^a^	3.597 ^bc^	0.333 ^d^	0.498 ^ab^	10.221 ^b^	45.146 ^b^	1.207 ^a^	23.996 ^fg^
		NS	1.559 ^a^	4.290 ^bc^	0.420 ^cd^	0.620 ^ab^	18.432 ^a^	42.703 ^b^	0.911 ^a^	43.916 ^bc^
	Means of landfills									
		BG	0.877 ^a^	4.022 ^b^	0.442 ^b^	0.627 ^a^	9.556 ^b^	45.363 ^a^	1.472 ^a^	30.281 ^b^
		NS	1.073 ^a^	5.933 ^a^	0.550 ^a^	0.666 ^a^	17.426 ^a^	38.309 ^a^	1.379 ^a^	46.653 ^a^
	Means of clones									
S	378		1.019 ^a^	6.796 ^a^	0.545 ^b^	0.571 ^a^	13.242 ^a^	35.989 ^b^	1.856 ^a^	47.750 ^b^
S	380		0.788 ^a^	4.405 ^ab^	0.718 ^a^	0.788 ^a^	13.597 ^a^	44.782 ^ab^	1.095 ^a^	58.631 ^a^
S	107/65/9		0.918 ^a^	5.414 ^ab^	0.505 ^bc^	0.681 ^a^	13.594 ^a^	39.266 ^ab^	1.465 ^a^	44.319 ^b^
P	135/81		1.045 ^a^	5.420 ^ab^	0.474 ^bc^	0.520 ^a^	12.359 ^a^	37.209 ^b^	1.392 ^a^	32.245 ^c^
P	I-214		0.946 ^a^	4.276 ^ab^	0.391 ^c^	0.516 ^a^	12.852 ^a^	31.564 ^b^	1.916 ^a^	24.161 ^d^
P	Pannonia		0.617 ^a^	4.655 ^ab^	0.441 ^bc^	0.475 ^a^	13.770 ^a^	38.542 ^b^	1.031 ^a^	32.532 ^c^
P	PE19/66		1.239 ^a^	4.909 ^ab^	0.516 ^bc^	1.061 ^a^	14.188 ^a^	63.411 ^a^	1.591 ^a^	34.140 ^c^
P	S1-8		1.226 ^a^	3.944 ^b^	0.376 ^c^	0.559 ^a^	14.326 ^a^	43.924 ^ab^	1.059 ^a^	33.956 ^c^

^(1)^—P-Poplar clones; S—Willow clones. ^(2)^—Location of landfill: NS—”Novi Sad” landfill; BG—”Vinča” landfill near Belgrade. ^(3)^ Chemical elements: As—arsenic; Ba—barium; Cd—cadmium; Cr—chromium; Cu—copper; Fe—iron; Ni—nickel; Zn—zinc. ^(4)^—Values with the same letter are not significantly different according to Duncan test for *p* < 0.05.

**Table 2 plants-14-01705-t002:** Duncan’s test for parameters of leaf gas exchange, biomass accumulation and the equivalent CO_2_ sequestration of the examined clones on “Vinča” and “Novi Sad” landfills.

P/S ^(1)^	Clone	Landfill ^(2)^	g_s_ ^(3)^	A	E	WUE	m_P_	CO_2 eq_
	Means of landfills							
		BG	35.296 ^b (4)^	2.573 ^b^	2.934 ^b^	0.857 ^b^	4315.810 ^b^	2125.536 ^b^
		NS	61.005 ^a^	4.713 ^a^	4.651 ^a^	1.019 ^a^	10,303.200 ^a^	5074.328 ^a^
	Means of clones							
S	378		54.722 ^abc^	5.450 ^a^	4.349 ^ab^	1.243 ^a^	8465.510 ^ab^	4169.265 ^ab^
S	380		48.600 ^bc^	3.333 ^b^	3.835 ^b^	0.916 ^bc^	5840.330 ^ab^	2876.364 ^ab^
S	107/65/9		59.067 ^ab^	4.713 ^a^	4.487 ^a^	1.016 ^ab^	5301.890 ^b^	2611.181 ^b^
P	135/81		55.222 ^abc^	3.075 ^bc^	4.331 ^ab^	0.716 ^c^	8046.670 ^ab^	3962.985 ^ab^
P	I-214		51.800 ^abc^	5.122 ^a^	4.046 ^ab^	1.208 ^a^	6823.560 ^ab^	3360.604 ^ab^
P	Pannonia		35.133 ^d^	2.444 ^c^	3.050 ^c^	0.735 ^c^	5528.030 ^ab^	2722.555 ^ab^
P	PE19/66		43.306 ^cd^	3.564 ^b^	3.688 ^bc^	0.909 ^bc^	5259.800 ^b^	2590.454 ^b^
P	S1-8		63.000 ^a^	3.544 ^b^	4.302 ^ab^	0.911 ^bc^	13,210.250 ^a^	6506.046 ^a^
	Means at the level of interaction clone x landfill							
S	378	BG	38.333 ^bcde^	3.189 ^bcde^	3.306 ^bcd^	0.966 ^abcd^	1915.390 ^c^	943.329 ^c^
S	378	NS	60.185 ^abc^	6.204 ^a^	4.697 ^ab^	1.335 ^a^	15,015.640 ^ab^	7395.201 ^ab^
S	380	BG	23.722 ^e^	2.056 ^de^	2.249 ^d^	0.961 ^bcd^	2214.840 ^c^	1090.811 ^c^
S	380	NS	65.185 ^ab^	4.185 ^b^	4.892 ^a^	0.887 ^cd^	9465.820 ^abc^	4661.917 ^abc^
S	107/65/9	BG	38.389 ^de^	1.978 ^de^	3.188 ^cd^	0.616 ^d^	2774.150 ^bc^	1366.271 ^bc^
S	107/65/9	NS	72.852 ^a^	6.537 ^a^	5.353 ^a^	1.283 ^ab^	7829.630 ^abc^	3856.091 ^abc^
S	135/81	BG	38.222 ^de^	2.600 ^cde^	3.409 ^c^	0.764 ^d^	3389.610 ^bc^	1669.382 ^bc^
S	135/81	NS	72.222 ^a^	3.550 ^bc^	5.253 ^a^	0.667 ^d^	12,703.730 ^abc^	6256.589 ^abc^
P	I-214	BG	33.500 ^de^	3.028 ^bcde^	2.879 ^cd^	0.949 ^bcd^	4676.170 ^abc^	2303.012 ^abc^
P	I-214	NS	64.000 ^ab^	6.519 ^a^	4.824 ^a^	1.380 ^a^	8970.950 ^abc^	4418.195 ^abc^
P	Pannonia	BG	27.722 ^de^	1.828 ^e^	2.326 ^d^	0.662 ^d^	5677.260 ^abc^	2796.052 ^abc^
P	Pannonia	NS	40.074 ^de^	2.856 ^cde^	3.533 ^c^	0.783 ^d^	5378.800 ^abc^	2649.058 ^abc^
P	PE19/66	BG	38.222 ^bcde^	2.378 ^bcde^	3.323 ^bcd^	0.749 ^cd^	4092.880 ^abc^	2015.744 ^abc^
P	PE19/66	NS	45.000 ^cd^	3.959 ^bc^	3.809 ^c^	0.962 ^cd^	6426.730 ^abc^	3165.164 ^abc^
P	S1-8	BG	47.941 ^bcd^	3.806 ^bc^	3.185 ^cd^	1.213 ^abc^	9786.160 ^abc^	4819.684 ^abc^
P	S1-8	NS	73.667 ^a^	3.358 ^bcd^	5.093 ^a^	0.698 ^d^	16,634.330 ^a^	8192.408 ^a^

^(1)^—P-poplar clones; S—Willow clones. ^(2)^—Location of landfill: NS—”Novi Sad” landfill; BG—”Vinča” landfill near Belgrade. ^(3)^—g_s_–stomatal conductance [mmol H_2_ O m^−2^ s^−1^]; A—net photosynthesis rate [µmol CO_2_ m^−2^ s^−1^]; E —transpiration rate [mmol H_2_ O m^−2^ s^−1^]; WUE—water use efficiency [mmol CO_2_ mol^−1^ H_2_ O]; m_P_—dry shoot biomass [kg ha^−1^]; CO_2 eq_—total accumulated CO_2_ equivalent [kg ha^−1^]. ^(4)^—Values with the same letter are not significantly different according to Duncan test for *p* < 0.05.

**Table 3 plants-14-01705-t003:** Physico-chemical properties of the soil from the “Novi Sad” landfill, near Novi Sad [19].

Granulometric Composition								
Horizon	Soil Depth (cm)	Coarse Sand (%)	Fine Sand (%)	Silt (%)	Clay (%)	Total Sand (%)	Total Clay (%)	Texture Class
SU1	0–10	24.34	58.46	11.76	5.44	82.80	17.20	Loamy Sand
SU2	10–40	15.67	68.17	10.04	6.12	83.84	16.16	Loamy Sand
SU3	40–100	37.53	43.67	9.00	9.80	81.20	18.80	Loamy Sand
P4	100–130	40.55	44.53	6.68	8.24	85.08	14.92	Loamy Sand
Chemical composition								
Horizon	Soil depth (cm)	CaCO_3_ (%)	pH	Humus (%)	Total nitrogen (%)	P_2_ O_5_ (mg/100 g)	K_2_ O (mg/100 g)	
SU1	0–10	1.40	7.50	1.90	0.031	4.98	3.86	
SU2	10–40	4.50	7.86	0.59	0.028	4.79	3.71	
SU3	40–100	2.64	7.63	0.43	0.035	5.27	4.10	
P4	100–130	6.11	7.54	0.32	0.025	4.57	3.52	

**Table 4 plants-14-01705-t004:** Physico-chemical properties of the soil from the “Vinča” landfill, near Belgrade [19].

Granulometric Composition								
Horizon	Soil Depth (cm)	Coarse Sand (%)	Fine Sand (%)	Silt (%)	Clay (%)	Total Sand (%)	Total Clay (%)	Texture Class
SU1	0–10	4.29	43.15	25.92	26.64	47.44	52.56	Sandy clay loam
SU2	10–40	4.29	41.67	26.96	27.08	45.96	54.04	Sandy clay loam
SU3	40–100	8.32	40.28	27.40	24.00	48.60	51.40	Sandy clay loam
Chemical composition								
Horizon	Soil depth (cm)	CaCO_3_ (%)	pH	Humus (%)	Total nitrogen (%)	P_2_ O_5_ (mg/100 g)	K_2_ O (mg/100 g)	
SU1	0–10	6.07	7.81	1.69	0.115	11.32	9.16	
SU2	10–40	2.09	7.71	1.67	0.118	11.58	9.38	
SU3	40–100	5.45	7.79	1.66	0.112	11.11	8.98	

**Table 5 plants-14-01705-t005:** Poplar and Willow Clones in Experiments Established at the “Vinča” and “Novi Sad” Landfills.

	Species	Clone	Registration Numbers
Poplar Clones	*Populus deltoides* Bartr. ex Marsh.	135/81	Experimental phase
		PE19/66	Experimental phase
		S1-8	7722/1
	*Populus x euramericana* (Dode) Guinier	I–214	Domesticated clone
		Pannonia	4/008-003/051
Willow clones	*Salix alba* L.	380	Experimental phase
		107/65/9	Experimental phase
		378	Experimental phase

## Data Availability

The raw data supporting the conclusions of this article will be made available by the authors on request.

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
