# Peer review of "Physiological Performance of Poplar and Willow Clones Growing on Metal-Contaminated Landfills"

_plants, 2025, doi:10.3390/plants14111705_

Round 1

Reviewer 1 Report

Comments and Suggestions for Authors

In this study, physiological responses of selected poplar and willow clones planted on two different landfill sites were evaluated, aiming to identify suitable clones for phytoremediation purposes at each landfill. However, several points need to be addressed:

1. The criteria used in this research for selecting poplar and willow clones were primarily based on physiological parameters such as net photosynthesis, transpiration, stomatal conductance, and water use efficiency, thus focusing exclusively on physiological vitality. The capability of these clones for contaminant uptake was not assessed. Although poplar and willow species are widely recognized for their potential in phytoremediation, verification of pollutant absorption capacities at the clonal level is necessary.

2. Although it is mentioned that irrigation was provided for plant growth at the "Vinča" landfill, specific details regarding irrigation conditions have not been clearly documented.

3. Physiological parameters such as net photosynthesis, transpiration, stomatal conductance, and water use efficiency were measured only once at each landfill site, specifically in July 2024. Similarly, height and diameter were recorded only once, in January and February of 2025. Given that poplar and willow species exhibit rapid growth, a more comprehensive, longer term dataset is required to reliably assess tree growth performance.

Author Response

Dear Reviewer,

Thank you very much for your time and effort in reviewing our manuscript. We sincerely appreciate your valuable and constructive comments.

We acknowledge that your suggestions were well-founded and helpful in improving the quality and clarity of our work. Most of the comments have been accepted and addressed accordingly in the revised version of the manuscript.

For transparency, we have included a detailed response to each comment in the attached document.

Thank you once again for your contribution to the improvement of our work.

Sincerely,

Lazar Kesić

Reviewer 2 Report

Comments and Suggestions for Authors

The manuscript is relatively good prepared, however give only small part of original results that are done only in Table 1. Other graphs (Fig. 1 and 2) are PCA analyses of measured parameters. Because this short study is compliant to previous published study of these authors (as they also mentioned it in the manuscript):

Kovačević, B., Milović, M., Kesić, L., Pajnik, L. P., Pekeč, S., Stanković, D., & Orlović, S. Interclonal Variation in Heavy Metal 455 Accumulation Among Poplar and Willow Clones: Implications for Phytoremediation of Contaminated Landfill Soils. Plants, 456 2025, 14(4), 567.

I recommend to publish it as Short Communication in Plants journal. However, due my knowledge this possibility is not offer to authors by Plants guidelines. For publishing as standard research article there is not enough minimal new data.

What is weak part of this manuscript, is not enough described Table 1 – Missing title for this table, what is BG and NS.

Due my understanding of values in Table 1, I do not think that (what is given at line 81): “clone 378 exhibited the lowest transpiration rate E”, because that are other clones that have lower one.

Because description of clones are at the end in Materials, I suggest to introduce Table 1 with including of information which clones are from Salix spp. and which from Populus spp. Also it will be welcome if authors cited their previous article and table in Materials here about description of soils and mention that they are heavy metal contaminated and how. If I understand correctly, the locality is same in both articles.

Other mistakes:

  • Please, delete residual bracket at line 31.
  • All Latin names must be written in italic font in the whole manuscript.
  • 107/65/9 at line 82 maybe?
  • Number 2 must be subscripted at line 111
  • Mistake at start of line 234
  • -1 as superscripted at line 312, 316

Author Response

Dear Reviewer,

We would like to sincerely thank you for your careful review and thoughtful comments on our manuscript.

Your observations were insightful and relevant, and we agree with the majority of the points you raised. We have revised the manuscript accordingly and incorporated most of your suggestions to improve the overall quality and clarity of the text.

For your reference, we have provided a point-by-point response to all comments in the attached file.

Thank you again for your valuable input and support in improving our work.

Sincerely, 

Lazar Kesić. 

Reviewer 3 Report

Comments and Suggestions for Authors

Dear All,

The manuscript evaluates the physiological responses and biomass production of poplar and willow clones cultivated on two landfill sites in Serbia, focusing on traits such as net photosynthesis, transpiration, stomatal conductance, and water use efficiency (WUE). The work explores clone × site interactions, physiological resilience, and carbon sequestration potential under varying edaphic conditions. Through robust field data, the authors distinguish between generalist and specialist clones with implications for phytoremediation and biomass stability in heterogeneous environments.

The research is highly relevant to the field of ecological restoration and sustainable land management. The approach of comparing clone performance in two distinct landfill settings provides original insights. By linking physiological performance with biomass yield and carbon equivalence, the study offers practical recommendations for landfill afforestation. Its scientific merit is strengthened by the use of gas exchange parameters and PCA analysis, although methodological depth and ecological contextualization require refinements.

Strong Points

  • The study investigates a timely and important ecological question using physiological and quantitative performance traits.

  • Field experiments across two degraded environments enhance ecological relevance.

  • The clone × environment interaction analysis and PCA are commendably applied.

  • Carbon sequestration data, contextualized by CO2 equivalents, enhances applied value.

  • The manuscript reflects a well-structured narrative and includes diverse clone profiles, providing generalist vs. specialist insights for phytoremediation strategies.

Weaker Aspects

In my opinion, the only handicapped points were:

  1. Rationale Development: The introduction insufficiently develops the rationale for clone selection or how specific site conditions (e.g., soil contamination) mechanistically impact physiological traits.

  2. Soil Characterization: Essential soil attributes (e.g., cation exchange capacity, micronutrients, EC, S-SO₄²⁻) were not included, which limits reproducibility and hinders deeper interpretations.

  3. Experimental Depth: The duration of the study is confined to a single growing season. Long-term validation is crucial for landfill-related afforestation.

  4. Terminology Consistency: Technical language inconsistencies such as "carbon accumulation" vs. "carbon equivalent" vs. "Ctot" should be standardized.

  5. Biomass Context: Comparison of biomass productivity to global averages lacks normalization (e.g., by precipitation, N inputs, or clone age).

  6. Water Dynamics: While groundwater access is discussed, no measurement of volumetric soil water or matric potential is included.

  7. Reference Incompleteness: Critical references on clone selection strategies, landfill ecotoxicology, and physiological trade-offs under multi-stress environments are missing.

  8. Graphical Presentation: Figures 1 and 2 lack units, PCA axis scaling, and legends explaining clone-site codes (e.g., “107/65-9 BG”).

  9. Statistical Significance Reporting: Statistical comparisons (Duncan’s test) are visually dense and could benefit from annotation in graphs or asterisks for significant clone × site contrasts.

So, I have annotated below (and along the manuscript) an attempt to clarify certain ideas, but the authors should examine my suggested wording changes carefully to be sure that I have not misinterpreted what they wanted to say.

Abstract

  • Page 1, Line 14: Replace "did not vary significantly" with “exhibited site-stable expression” to enhance scientific specificity.

  • Page 1, Line 22: Avoid “gas exchange” alone – specify “leaf-level gas exchange traits (A, E, gs)”.

Introduction

  • Page 2, Line 43: The sentence “Many of these contaminants pose substantial threats…” could integrate quantification (e.g., ppm levels of Zn, Cd, etc.).

  • Page 2, Line 68: Reframe aim with a hypothesis: "We hypothesize that clone-specific physiological adaptation to landfill conditions will differentially influence biomass production depending on water access and soil fertility."

Methods

  • Page 9, Line 267: The mention of groundwater access depth lacks actual water table measurements or soil water retention parameters.

  • Page 10, Line 296: Please include clone registration numbers for transparency.

  • Page 11, Line 327: Photosynthetic gas exchange was taken from single leaves – replicate leaves per plant or canopy corrections would improve rigor.

Results and Discussion

  • Page 4, Line 97: Please indicate biomass yield standard error or confidence intervals.

  • Page 6, Line 124–128: PCA loadings are said to be negative across traits – clarify the axis scale and biological interpretation.

  • Page 7, Line 216: The expression “high water consumption” is vague – better: “increased transpiration efficiency under non-limiting conditions”.

  • Page 7, Line 221: Discuss why S1-8 maintained high gs and E yet had moderate WUE – possible trade-offs?

Conclusion

  • Page 11, Line 356: The sentence “dominant effect of transpiration…” could be reframed: "Under abundant water availability, gas exchange traits (gs, E, A) were stronger predictors of biomass yield than intrinsic water use efficiency (WUEi)."

Author Response

Dear Reviewer,

Thank you very much for your constructive and helpful review of our manuscript.

We found your comments to be well-justified and have taken them into careful consideration during the revision process. Most of your suggestions have been implemented, and we believe they have significantly improved the quality of the manuscript.

A detailed, point-by-point response to your comments is provided in the attached file.

We appreciate your time and contribution to the refinement of our work.

Sincerely,

Lazar Kesić. 

Round 2

Reviewer 2 Report

Comments and Suggestions for Authors

The manuscript was fulfilled about interesting values of metal accumulations in plants and now, I can recommend publishing it. However, before that it is important to specify of which “aboveground biomass” was used for metal determination. If it was leaves, wood…?

Also, it is very important to check all values in Table 1 if they are correctly written in English – if these values are in thousands (comma) or decimal numbers (decimal dot).

It is necessary to specify the methodology for metal determination together with description of preparation of samples for this determination. Did metals determine by ICP, AAS or other method?

Author Response

Dear Reviewer,

Thank you very much for your positive evaluation and recommendation for publication. I truly appreciate your time and effort in reviewing the manuscript.

Please find my responses to your comments and suggestions in the attached document.

Sincerely,
On behalf of all authors

Reviewer 3 Report

Comments and Suggestions for Authors

Dear Editors,

The revised manuscript entitled “Physiological performance of poplar and willow clones on landfills” has been substantially improved. The authors have thoughtfully addressed the critical comments and suggestions raised during the prior round of peer review.

Notably, the clarification of clone × site interactions, the elaboration on physiological trait implications, and the refinement of biomass productivity interpretation significantly enhance the scientific merit of the work. The newly added details on methodological consistency, PCA loadings, and physiological trade-offs, particularly for clone S1-8, demonstrate a commendable effort by the authors to strengthen the ecological interpretation and reproducibility of their findings.

The manuscript now presents a more robust discussion of clone-specific adaptability under landfill conditions, successfully balancing applied phytoremediation objectives with mechanistic insights. While minor limitations remain (e.g., multi-season validation and hydrological profiling), they do not detract from the study’s overall quality, field relevance, and publication suitability.

I therefore support the acceptance of this revised manuscript in its current form.

Kind regards,

JLJ

Author Response

Dear Reviewer,

We sincerely thank you for your careful reading of the revised manuscript and for the highly constructive comments provided during the previous stages of the review process, which significantly contributed to the improvement of our work. We greatly appreciate your positive remarks regarding the clarifications we made on the clone × site interactions, the interpretation of physiological traits and biomass productivity, as well as the added details concerning methodological consistency, PCA analysis, and physiological trade-offs, particularly for clone S1-8.

We also thank you for pointing out the remaining minor limitations (such as multi-season validation and hydrological profiling), which we acknowledge and plan to address in future research.

Once again, thank you for your time, expertise, and your recommendation for the acceptance of the manuscript.

Sincerely,

on behalf of the all authors.